

# 1 Quantifying TOLNet Ozone Lidar Accuracy during the 2014
# 2 DISCOVER-AQ and FRAPPÉ Campaigns

Lihua Wang[1], Michael J. Newchurch[1], Raul J. Alvarez II[2], Timothy A. Berkoff[3], Steven S.
Brown[2], William Carrion[3,4], Russell J. De Young[3], Bryan J. Johnson[2], Rene Ganoe[4], Guillaume
Gronoff[3,4], Guillaume Kirgis[2,5], Shi Kuang[1], Andrew O. Langford[2], Thierry Leblanc[6], Erin E.
McDuffie[2,5,7], Thomas J. McGee[8], Denis Pliutau[4], Christoph J. Senff[2,5], John T. Sullivan[8], Grant
Sumnicht[4], Laurence W. Twigg[4], Andrew J. Weinheimer[9]
[1]University of Alabama in Huntsville, Huntsville, Alabama, USA
[2]NOAA Earth System Research Laboratory, Boulder, Colorado, USA
[3]NASA Langley Research Center, Hampton, Virginia, USA
[4]Science Systems and Applications Inc., Lanham, Maryland, USA
[5]Cooperative Institute for Research in Environmental Sciences, University of Colorado, Boulder, Colorado, USA
[6]Jet Propulsion Laboratory, California Institute of Technology, Wrightwood, California, USA
[7]Department of Chemistry, University of Colorado, Boulder, Colorado, USA
[8]NASA Goddard Space Flight Center, Greenbelt, Maryland, USA
[9]National Center for Atmospheric Research, Boulder, USA
Correspondence to Shi Kuang (kuang@nsstc.uah.edu)



**Abstract**
The Tropospheric Ozone Lidar Network (TOLNet) is a unique network of lidar systems that measure high-
resolution atmospheric profiles of ozone. The accurate characterization of these lidars is necessary to determine the
uniformity of cross-instrument calibration. From July to August 2014, three lidars, the TROPospheric OZone
(TROPOZ) lidar, the Tunable Optical Profiler for Aerosol and oZone (TOPAZ) lidar, and the Langley Mobile
Ozone Lidar (LMOL), of TOLNet participated in the "Deriving Information on Surface conditions from Column
and Vertically Resolved Observations Relevant to Air Quality" (DISCOVER-AQ) mission and the "Front Range Air
Pollution and Photochemistry Éxperiment" (FRAPPÉ) to measure ozone variations from the boundary layer to the
top of the troposphere. This study presents the analysis of the intercomparison between the TROPOZ, TOPAZ, and
LMOL lidars, along with comparisons between the lidars and other *in situ* ozone instruments including ozonesondes
and a P-3B airborne chemiluminescence sensor. In terms of the range-resolving capability, the TOLNet lidars
measured vertical ozone structures with an accuracy generally better than ±15% within the troposphere. Larger
differences occur at some individual altitudes in both the near-field and far-field range of the lidar systems, largely
as expected. In terms of column average, the TOLNet lidars measured ozone with an accuracy better than ±5% for
both the intercomparison between the lidars and between the lidars and other instruments. These results indicate
very good measurement accuracy for these three TOLNet lidars, making them suitable for use in air quality, satellite
validation, and ozone modeling efforts.
**1.    Introduction**
**1.1 TOLNet**
The Tropospheric Ozone Lidar Network (TOLNet) provides time-height measurements of ozone from the
planetary boundary layer (PBL) to the top of the troposphere at multiple locations for satellite validation, model
evaluation, and scientific research (Newchurch et al., 2016; http://www-air.larc.nasa.gov/missions/TOLNet/).
Particularly, these high-fidelity ozone measurements can serve to validate NASA's first Earth Venture Instrument
mission, Tropospheric Emissions: Monitoring Pollution (TEMPO), planned to launch in 2019. A second objective of
TOLNet is to identify a brassboard ozone lidar instrument that would be suitable to populate a network to address an
increasing desire for ozone profiles by air-quality scientists and managers within the modeling and satellite
communities (Bowman, 2013).
TOLNet consists of five ozone lidars across the United States and one in Canada: the Table Mountain
tropospheric ozone differential absorption lidar (DIAL) at NASA's Jet Propulsion Laboratory, the Tunable Optical
Profiler for Aerosol and oZone (TOPAZ) lidar at NOAA's Earth System Research Laboratory (ESRL), the Rocket-
city Ozone ($O_3$) Quality Evaluation in the Troposphere ($RO_3QET$) lidar at the University of Alabama in Huntsville
(UAH), the TROPospheric OZone (TROPOZ) DIAL at NASA's Goddard Space Flight Space Center (GSFC), the
Langley Mobile Ozone Lidar (LMOL) at NASA's Langley Research Center (LaRC), and Autonomous Mobile
Ozone Lidar Instrument for Tropospheric Experiments (AMOLITE) at Environment and Climate Change Canada.
All TOLNet lidars have unique configurations that are associated with their original measurement design
purposes, including their transmitter, receiver, and signal processing systems. Most components of these lidars are
customized and differ significantly in pulse energy, repetition rate, receiver size, solar (or narrow-band) interference
filter, and range resolution. These differences result in varying signal-to-noise ratios (SNRs), which impact the
useful operating ranges and statistical uncertainties in ozone retrieval. The selection of the DIAL wavelengths
determines the sensitivity to interference by other species, primarily aerosols. In addition, multiple lidar data
processing and retrieval algorithms could also lead to different effective resolutions and lidar retrieval uncertainties
(Godin et al., 1999; Leblanc et al., 2016). Therefore, it is important to quantify the measurement differences between
the TOLNet lidars and understand their sources before we form a consistent TOLNet dataset. A previous
intercomparison between TROPOZ and LMOL reported by Sullivan et al. (2015) concluded that the observed ozone
column averages from the two lidars were within ±8% of each other, and their ozone profiles were mostly within
±10% of each other. That particular study served as the first reported measurement intercomparison of two ground-
based tropospheric ozone lidar systems within the United States.
**1.2 DISCOVER-AQ 2014 and FRAPPÉ Campaigns**
The scientific goal of the TOLNet lidars in this study was to provide continuous, high-resolution
tropospheric ozone profiles to support the NASA-sponsored DISCOVER-AQ mission
(https://www.nasa.gov/larc/2014-discoveraq-campaign/), and the National Science Foundation (NSF) and state of
Colorado (CO) jointly sponsored FRAPPÉ (Dingle et al., 2016) from July to August 2014. By collaborating with
FRAPPÉ, the 2014 CO study was the final stop in a series of four field campaigns by DISCOVER-AQ to understand
sources, transport and chemical transformations of air pollutants, particularly those that lead to ground-level ozone
formation (Crawford and Pickering, 2014).
Prior to the two campaigns, TOPAZ, TROPOZ, and LMOL were all deployed to the same location in Erie,
CO to obtain intercomparison data at the Boulder Atmospheric Observatory (BAO) (40.050°N, 105.003°W, 1584 m
above sea level, ASL). Subsequent to the BAO intercomparison, TROPOZ and LMOL re-deployed to locations near
Fort Collins, CO (~60 km north-northwest of BAO) and Golden, CO (~40 km southwest of BAO), respectively, for
their different scientific missions. During the DISCOVER-AQ and FRAPPÉ campaigns, balloon-borne ozonesondes
were launched at selective sites. In addition, the NASA P-3B aircraft performed multiple spiral ascents and descents
over several ground sites and provided numerous vertical profiles of ozone measurements. In this study, we compare
retrievals between the three lidars and evaluate the ozone lidar accuracy using ozonesonde and P-3B aircraft
measurements. These two campaigns offered a unique opportunity for the lidar validation work, as they involved so
many different instruments.
**2.  Instruments**
**2.1 TOLNet Lidars**




**Table 1** lists the main hardware specifications of the three TOLNet lidars and their ozone retrieval
processes, which could potentially impact the intercomparison result.

### 2.1.1 TROPOZ/NASA GSFC

The transmitter for TROPOZ consists of two 50-Hz Nd:YAG- lasers used to pump two Raman cells filled
with Deuterium ($D_2$) and Hydrogen ($H_2$) gases, respectively, to generate two outgoing lasers at 289 and 299 nm. The
typical pulse energies are 12 mJ at 299 nm (off-line) and 16 mJ at 289 nm (on-line) (Sullivan et al., 2014). The
receiving system consists of a 45-cm-diameter Newtonian telescope for measuring far field and four smaller 2.5-cm
refracting telescopes to measure near field. The 45-cm telescope has a 1-mrad field of view (FOV), and the 2.5-cm
telescopes have a much wider FOV at 10 mrad. In each channel, solar interference filters with a 1-nm bandwidth
decrease the amount of ambient solar light, which improves the SNR. The fundamental range resolution for the data
acquisition system is 15 m (100 ns). TROPOZ measures ozone up to 16 km during daytime hours and higher
altitudes at night.

### 2.1.2 TOPAZ/NOAA ESRL

The TOPAZ lidar is a truck-mounted zenith-looking, scanning instrument modified from the nadir-looking
airborne DIAL configuration first used in the 2006 Texas Air Quality Study (TexAQS II) (Alvarez et al., 2011;
Senff et al., 2010). The lidar transmitter is based on a Ce:LiCAF laser pumped by a quadrupled Nd:YLF laser to
produce three UV wavelengths, each at a 333 Hz repetition rate and tunable from 283 nm to 310 nm. The actual
wavelengths used during DISCOVER-AQ 2014 were 287, 291, and 294 nm. Compared to the conventional two-
wavelength DIAL, the three-wavelength configuration can potentially minimize the aerosol interference by using the
dual-DIAL retrieval technique (Kovalev and Bristow, 1996) without assuming a lidar ratio and Angström exponent.
However, in this study, ozone was retrieved using the 287- and 294-nm lidar signals and the standard two-
wavelength DIAL algorithm because the two-wavelength retrieval was less affected by significant lidar signal noise
(Alvarez et al., 2011).
Laser light backscattered by air molecules and aerosol particles is collected with a co-axial 50-cm diameter
Newtonian telescope and then split at a 1:9 ratio into near- and far-field detection channels. The FOVs of the near-
and far-field channels are controlled by different-size apertures resulting in full overlap at distances of ~300 m and
~800 m, respectively. Both channels use gated photomultipliers (PMTs) operated in analog mode with solar
interference filters during the daytime. Compared to photon counting (PC) signals, the analog signal is able to keep
high linearity for strong signals and is particularly suitable for near-range measurement. The two-axis scanner on the
truck permits pointing the laser beam at several shallow elevation angles at a fixed, but changeable azimuth angle,
typically at $2^o$, $6^o$, $20^o$, and $90^o$ elevation angles that are repeated approximately every 5 minutes. The ozone profiles
at these four angles are spliced together to create composite vertical profiles extending from 10 m to about 2 km
AGL (Langford et al., 2016). The range resolution of the signal recording system is 6 m.
During the 2014 DISCOVER-AQ and FRAPPÉ campaigns, the TOPAZ ozone observations at low
elevation angles ($2^o$, $6^o$, and $20^o$) suffered from a slight, but consistent range-dependent bias created by an unknown





source of noise in the data acquisition system. The cause of this noise remains unknown and attempts to correct the
resulting bias were unsuccessful. This bias manifests itself primarily in the low-angle observations because the
signal levels and SNR are significantly lower compared to the measurements at 90°. For these reasons, the low angle
observations below 500 m were excluded from the comparisons reported within this study.
**2.1.3 LMOL/NASA LaRC**
The transmitter of LMOL consists of a diode-pumped Nd:YLF laser pumping a Ce:LiCAF tunable UV
laser to obtain two wavelengths typically at 287.1 and 292.7 nm with a pulse energy of 0.2 mJ at 500 Hz for each
wavelength. The lidar receiver system consists of a 40-cm telescope with a 1.4-mrad FOV to measure far field and
another 30-cm telescope with an adjustable FOV to measure near field (De Young et al., 2017). The raw lidar
signals are recorded with a 7.5-m range resolution. The LMOL data acquisition system operates in both analog and
PC modes. In this study, LMOL measures ozone between 0.7 and 4.5 km. Ozone measurements for DISCOVER-AQ
represent LMOL's very first remote deployment.
**2.1.4 Lidar Data Processing and Retrieval Algorithms**
The data processing and DIAL retrieval algorithms for the three TOLNet lidars are similar but not identical.
Their details have been described by Alvarez et al. (2011), De Young et al. (2017), Langford et al. (2011), and
Sullivan et al. (2015; 2014). Some basic procedures were applied on the raw lidar signals before retrievals, such as
time integration (5 min for this study), dead-time correction (for PC only), background correction, merging of PC
and analog signals (for a system with both PC and analog channels), and signal-induced-bias (SIB) correction
(Kuang et al., 2013). Some parameters are system dependent or empirical due to different equipment, such as the
dead-time value, PC-analog timing offset, averaging range for background calculation, and SIB simulation function.
All groups agreed to use the Brion-Daumont-Malicet (BDM) database (Daumont et al., 1992; Malicet et al., 1995;
Brion et al., 1993) to calculate differential ozone absorption cross-sections, which are temperature-dependent.
The ozone number density profile results from computing the derivative of the logarithm of the on-line to
off-line signal ratios. Spatial smoothing is usually necessary to improve the SNR and reduce the statistical errors.
Various smoothing methods and their impacts on final lidar retrieval have been described by Godin et al. (1999).
Both TROPOZ and LMOL groups applied a Savitzky-Golay (SG) filter with a 2$^{nd}$ degree polynomial on the
derivative of the logarithm of the on-line to off-line signal ratios with an increasing window width to accommodate
the quickly decreasing SNR. However, the SG window sizes for TROPOZ and LMOL are different due to different
SNRs at each altitude. The TOPAZ group smoothed the derivative with a five-point least-square fitting in a 450-m
interval. The different retrieval methodologies and parameters affect the effective vertical resolution of the retrieved
ozone profiles, as listed in Table 1. This effective resolution determines the capability of the lidars to resolve vertical
ozone structure and is not equal to, but is associated with, the fitting window width.
All groups applied similar schemes to correct the aerosol interference. These schemes iteratively substitute
derived ozone from the DIAL equation into the lidar equation to solve aerosol extinction and backscatter until both
aerosol and ozone converge (Alvarez et al., 2011; Kuang et al., 2011; Sullivan et al., 2014). The differential aerosol



backscatter and extinction were calculated with the approximation from Browell et al. (1985). Lidars directly
measure the ozone number density, and all three groups used the same temperature and pressure profiles from co-
located ozonesonde measurements for Rayleigh correction, ozone mixing-ratio calculations, and computation of the
temperature dependent ozone absorption cross sections.
Merging between different altitude channels, either different telescopes or different optical channels of the
same telescope, is challenging with limited methodologies reported in the literature (Kuang et al., 2011). It is
difficult to specify a method for all groups because merging is system-dependent and is affected by many factors
previously described. Therefore, the three lidar groups merge the ozone profiles at different altitudes optimized for
their system and SNR levels such as the example method described by Sullivan et al. (2015). As a result, additional
differences between systems can occur due to the non-standardized altitude channel merging.
**2.1.5 Error budget of the lidar measurements**
Only a brief description of the error budget of the lidar measurements is provided in this paper since the
details have been discussed in the respective instrument paper (Alvarez et al., 2011; De Young et al., 2017; Sullivan
et al., 2014). Table 2 presents the estimated measurement uncertainties for 5 or 30-min integration time for the three
lidars. Statistical errors (Papayannis et al., 1990) arising from signal and background noise fluctuations are random
errors and may be improved by additional averaging or smoothing. The maximum statistical uncertainties for the
three lidars are similar (20% for 5 min and 8% for 30 min) within their measurable ranges although they are
different at the same altitude. The uncertainty arising from aerosol interference could be the largest systematic error
source and can be minimized by using the appropriate correction algorithm (Eisele and Trickl, 2005; Immler, 2003;
Sullivan et al., 2014). The estimated total lidar measurement uncertainties are 22% and 13% for 5 and 30 min,
respectively, within the lidar measurement ranges listed in Table 1.
**2.2 Ozonesondes**
An ozonesonde is a lightweight, balloon-borne instrument that consists of a Teflon air pump and an ozone
sensor interfaced to a meteorological radiosonde. The ozone sensor uses an electrode electrochemical cell containing
potassium iodide (KI) solution (Komhyr, 1969; Komhyr et al., 1995) to measure ozone with a precision better than
±5% and an accuracy better than ±10% up to 35 km altitude with a sampling interval of about 1 s and a retrieval
vertical resolution of 100 m (Deshler et al., 2008; Johnson et al., 2008; Smit et al., 2007). The uncertainty of
ozonesonde measurement is larger in the troposphere than that in the stratosphere (Liu et al., 2009). As the balloon
carrying the instrument package ascends through the atmosphere, the pump bubbles ambient air into the sensor cell.
The reaction of ozone and iodide generates an electrical signal proportional to the amount of ozone. A radiosonde
attached in the same package measures air temperature, pressure, and relative humidity (Stauffer et al., 2014).
Ozonesondes are capable of measuring ozone under various weather conditions (e.g., cloudy, thunderstorm). The
free-flying ozonesondes typically reach 35-km altitude in less than two hours with a rise rate at about 5 m/s.





**2.3 Ozone Measurement Instrument onboard NASA's P-3B**

NASA's P-3B aircraft is a pressurized, four-engine turboprop, capable of long-duration flights of 8-12 hours and is based out of NASA's Wallops Flight Facility in Wallops Island, Virginia. A series of gas and aerosol instruments were outfitted within the P-3B aircraft. Ozone was measured using the National Center for Atmospheric Research (NCAR)'s 4-channel chemiluminescence instrument based on the reaction between ambient ozone and nitric oxide (NO) with an accuracy of about ±5% and sampling interval of 1 s (Weinheimer et al., 1993; Ridley et al., 1992). The precision of this ozone detector is better than ±1% when ambient ozone is higher than 10 ppbv. The P-3B aircraft flew spirals from 300 m to 4570 m above the surface over selected ground monitoring sites including all three lidar sites (more information in Section 3.3) during the DISCOVER-AQ 2014 campaign.

**3.  Results**

**3.1  Lidar Intercomparisons**

The three TOLNet lidars were deployed next to the BAO tower to take simultaneous measurements before the DISCOVER-AQ/FRAPPÉ campaign. They were only a few hundreds of meters away from each other and were within 5 m of the same elevation (see measurement locations in Table 1).

Unlike stratospheric ozone lidars that focus on integrating hours of observations, tropospheric ozone lidars need to detect ozone variations with timescales on the order of minutes, when considering ozone's shorter lifetime, smaller-scale transport, and mixing processes within the PBL and free troposphere (Steinbrecht et al., 2009; McDermid et al., 1990). Therefore, we processed all lidar data on a 5-min temporal scale (signal integration time). Rayleigh correction was performed with the same atmospheric profile from the ozonesonde. Because the three lidars have different fundamental range resolutions, retrieved ozone number density values were internally interpolated on the same altitude grid with a 15-m interval for comparison.

Figure 1 presents the comparison of the TOPAZ and TROPOZ observed ozone at BAO from 1300 to 2135 UTC (6 hours ahead of local time, Mountain Daylight Time) on July 11, 2014 under a partly cloudy sky condition. Data influenced by cloud interferences were filtered out. Ozone curtains from both lidars (Figure 1 a and b) show a significant (about 40%) ozone increase in the early afternoon. A total of 7655 TOPAZ and TROPOZ coincident pairs were constructed between 0.6 and 2 km AGL (altitude range over which both lidars provided valid data) over this time period. The measurement differences between the two lidars are mostly within ±5% at individual grids (Figure 1 c). The product of averaged ozone concentration over some specified altitude range can represent the atmospheric ozone abundance and can be also useful for satellite validation. Here, we refer this product as ozone column average with the unit of number density, not to be confused with integrated column ozone often reported in Dobson units. The statistics of the intercomparison of the column averages is listed in Table 3. The similar 1σ standard deviations (17.8 and 16.7 x $10^{16}$ molec·m$^{-3}$) suggest similar ozone variations captured by both lidars. The mean relative difference (or normalized bias) was calculated by averaging the relative difference (i.e., (TROPOZ-TOPAZ)/TOPAZ, the denominator was arbitrarily chosen) for all paired ozone profiles. The -1.1±2.6% mean



relative difference suggests excellent agreement of the averaged ozone column (Figure 1 d) for 80 profiles over 6.5
hours between TOPAZ and TROPOZ retrievals.

Figure 2 shows the TOPAZ-LMOL intercomparison for data taken on July 16, 2014 with 1902 coincident

pairs from 0.9 to 2 km and between 1340 to 1730 UTC on this day. Some of the data gaps were due to low clouds
blocking the lidar beams. The retrievals between the two lidars agree with each other mostly within ±10% (Figure 2
c). LMOL measured a mean ozone column average (Figure 2 d) 3.8±2.9% lower than TOPAZ for a total of 28
paired profiles, which is significantly fewer than those from the TROPOZ-TOPAZ comparison.

The generally random distribution of the relative differences in Figure 1 (c) and 2 (c) suggests overall

consistent measurements with small systematic errors from all three lidars. In summary, TROPOZ, LMOL, and
TOPAZ report ozone values at individual altitudes mostly within ±10%, which is well within their respective
uncertainties and report ozone column averages within ±3.8% on average.
**3.2    Lidars versus Ozonesondes**

In order to compare the lidar data to ozonesondes, the Rayleigh- and aerosol-corrected lidar data was

converted from ozone number densities to ozone mixing ratios by using sonde-measured pressure and temperature
profiles, and averaged over a 30-minute interval (±15 minutes around sonde launch times). The ozonesondes report
values approximately every second (about every 5 m in altitude) in raw data. For comparison, the ozonesonde raw
data were linearly interpolated on the lidar altitude grids with a 15-meter interval. Figure 3 shows the mean ozone
mixing ratios measured by TOLNet lidars and ozonesondes, as well as their mean relative difference as function of
altitude.

After the DISCOVER-AQ/FRAPPÉ campaign started, the TROPOZ lidar deployed to Fort Collins, CO to

measure ozone. There were 11 ozonesonde profiles that were coincident and co-located with the TROPOZ
measurements. The mean ozone profiles of TROPOZ and sondes (Figure 3a) show similar vertical variations with
enhanced PBL and upper tropospheric ozone. The mean relative differences between TROPOZ and ozonesondes
(Figure 3b) are mostly within ±10% up to 9 km. The local maximum of the differences at 1.8 km is associated with
the merging of ozone retrievals from the near-field channel and far-field channel. Above 9 km, the biases start to
increase and exceed 25% with large oscillations due to large statistical errors as a consequence of low SNR. Biases
between 10-20% are still very representative of the upper free troposphere. On average for altitudes from 0.35 to 12
km, TROPOZ measures 2.9% higher ozone than the ozonesondes. This difference can be seen as the mean
difference of ozone column average between the ozonesondes and lidar for a 30-min integration time.

Between July 10 and July 16, a total of 10 ozonesondes were released near the BAO tower and 7 of them

were coincident with TOPAZ measurements (3 on July 10, 3 on July 11, and 1 on July 16). TOPAZ mostly agrees
with ozonesondes between -5% and 10% (Figure 3 c, d). Compared to ozonesondes, TOPAZ measures 4.4% more
PBL ozone on average.

On July 16, there was only one pair of coincident LMOL and ozonesonde measurements at the BAO tower

(Figure 3 e, f). The 30-minute averaged LMOL ozone profile agrees with the ozonesonde mostly within 0-15%





between 0.95 and 4.5 km AGL with an overall average of 6.2%**.** The maximum bias occurring at far range (above 4
km) is principally due to low SNR. The bias observed at 1.5 km is likely due to the high variation in aerosol
concentration, that was also observed in the green channel. Since there is only one comparison between the LMOL
and ozonesonde, the statistical information on the overall bias between their measurements is not available.

In summary, all three TOLNet lidars exhibit overall positive bias, up to 4.4%**,** compared to ozonesondes

excluding the single profile comparison to  LMOL (6.2%). The larger bias than the climatological difference
between lidar and ozonesondes reported by Gaudel et al. (2015) (0.6 ppbv) could be associated with the much
shorter averaging time period. The maximum biases exist in two regions, near-range altitudes and far-range
altitudes. The large far-range bias is expected and is primarily associated with the high statistical errors arising from
low SNR. The large near-range bias is more complicated and could be associated with various factors, primarily the
aerosol correction and the merging of signal or ozone from different optical or altitude channels.
**3.3       Lidars versus P-3B Chemiluminescence Instrument**

During the campaigns, the P-3B aircraft measured ozone profiles while doing spirals above the lidar sites.

There are 34 coincident profiles between TROPOZ and the P-3B at Fort Collins, 29 between TOPAZ and the P-3B
at the BAO tower, and 9 between LMOL and the P-3B at Golden, CO. The distances between the lidar and P-3B
spiral centers for these paired profiles were less than 11 km. To make coincident pairs between P-3B and lidar data,
we interpolate the P-3B data onto the lidar vertical grids with a 15-m vertical resolution. Figure 4 shows the average
ozone profiles measured by the lidars and the P-3B as well as their mean relative differences. TROPOZ and the P-
3B agree with each other within ±5% between 0.5 to 3.5 km (Figure 4 a, b) with a -0.8% overall average relative
difference. TOPAZ agrees with the P-3B within -11% and 3% between 0.5 and 2 km (Figure 4 c, d) with a **-**2.7%
overall average relative difference. TOPAZ underestimates the lower-PBL (<1.5 km) ozone compared to P-3B, but
when compared to ozonesondes TOPAZ overestimates ozone at many of these same altitudes (see Figure 3 d).
LMOL agrees with P-3B mostly within -5% and 0% above 1800 m and within -15% and -5% between 0.7-1.8 km
(Figure 4 e, f) with a -4.9% overall average relative difference.

In summary, TOPAZ and LMOL exhibited noticeable negative bias in the PBL compared to the P-3B while

TROPOZ measured slightly lower than the P-3B. The differences between the two lidars and the P-3B are not
significantly correlated suggesting that the problem was not likely from the P-3B ozone instrument. These
differences could at least in part be caused by the lidar systematic errors mentioned in Section 2.1.5, but could also
reflect horizontal ozone variability across the P-3B spirals, which were up to 22 km in diameter.
**4.       Summary and Conclusions**

Intercomparisons have been made between three of the six TOLNet ozone lidars (NASA GSFC's

TROPOZ, NOAA ESRL's TOPAZ, and NASA LaRC's LMOL) and between the lidars and other *in situ* ozone
measurement instruments using coincident data during the 2014 DISCOVER-AQ and FRAPPÉ campaigns at
NOAA's BAO in Erie, CO. On average, TROPOZ, TOPAZ, and LMOL reported very similar ozone within their
reported uncertainties for a 5-min signal integration time. The three lidars measured consistent ozone variations



revealed in the lidar time-height curtains and in the distribution of their relative differences. From intercomparisons
between the lidars and other instruments we find **(1)** All lidars measure higher ozone than ozonesondes with an
averaged relative difference within 4.4%. The lidar profile measurements agree with the ozonesonde observations
within -10-15% in their measurable ranges except at a few near-field altitudes. These results are generally consistent
with Sullivan et al. (2015) from a similar ozonesonde-lidar intercomparison. **(2)** TROPOZ agrees with the P-3B
chemiluminescence Instrument below 3.5 km within ±5% with a small column-averaged relative difference of -
0.8%. TOPAZ and LMOL exhibit a slightly larger bias mostly between -15% and 5% below 2 km compared to P-3B
with a column-averaged difference of -2.7% and -4.9%, respectively.
Overall, intercomparisons between themselves and with *in situ* instruments suggest that the TOLNet lidars
are capable of capturing high-temporal tropospheric-ozone variability and measuring tropospheric ozone with
accuracy better than ±15% in terms of their vertical resolving capability and better than ±5% in terms of their
column measurement. These lidars have sufficient accuracy for model evaluation and satellite validation (Liu et al.,
2010). Since the 2014 campaigns, the TOLNET lidars have been modified to improve their stability and their
accuracy. The validation of these upgraded lidars will be reported in a future paper.
**Acknowledgement**
This work is supported by the TOLNet program developed by the National Aeronautics and Space
Administration (NASA)'s Science Mission Directorate and by the National Oceanic and Atmospheric
Administration Earth System Research Laboratory. Dr. John T. Sullivan's research was supported by an
appointment to the NASA Postdoctoral Program at the NASA Goddard Space Flight Center, administered by
Universities Space Research Association under contract with NASA. The views, opinions, and findings contained in
this report are those of the authors and should not be construed as an official NOAA, NASA, or U.S. Government
position, policy, or decision.



**Table 1. Specifications for the TOLNet lidars.**

|  | TROPOZ | TOPAZ | LMOL |
|---|---|---|---|
| **Transmitter** | | | |
| Laser type | Nd:YAG pumped $D_2$, $H_2$ Raman cell | Nd:YLF pumped Ce:LiCAF | Nd:YLF pumped Ce:LiCAF |
| Wavelengths (nm) | 288.9, 299.1 | 287, 291, 294 | 287.1, 292.7 |
| Pulse Repetition Rate (Hz) | 50 | 333 | 500 |
| Pulse energy (mJ) | 12 (299 nm), 16 (289 nm) | ~0.06 for all wavelengths | 0.2 for both wavelengths |
| **Detection and data acquisition system** | | | |
| Telescope diameter (cm) | 45, 2.5 | 50 | 40, 30 |
| FOV (mrad) | 1 (45 cm), 10 (2.5 cm) | 1.5 (far field channel), 3 (near field channel) | 1.4 (far field channel), variable FOV (near field channel) |
| Signal detection type | PMT | PMT | PMT |
| Data acquisition type | PC | Analog | Analog and PC |
| Fundamental range resolution (m) | 15 | 6 | 7.5 |
| Instrument reference | (Sullivan et al., 2014) | (Alvarez et al., 2011) | (DeYoung et al., 2017) |
| **DIAL retrieval** | | | |
| DIAL retrieval and smoothing method | $1^{st}$-order (differential) SG filter with a $2^{nd}$ degree polynomial with an increasing window width applied on the derivative of the logarithm of the signal ratios | five-point least square fitting with a 450-m window applied on the derivative of the logarithm of the signal ratios | $1^{st}$-order (differential) SG filter with a $2^{nd}$ degree polynomial, with an increasing window width applied on the derivative of the logarithm of the signal ratios |
| Retrieval effective resolution (m) | ~100 at 1 km degrading to ~800 at 10 km | ~10 below 50 m, ~30 from 50 to 150 m, ~100 from 150 to 500 m, 315 above 500 m | 225 below 3 km degrading to 506 above 3 km |
| Aerosol correction reference | (Kuang et al., 2011; Sullivan et al., 2014) | (Alvarez et al., 2011) | (Browell et al., 1985; DeYoung et al., 2017) |
| Valid altitudes (km above ground level, AGL) | 0.35-16 | 0.01-2 | 0.7-4.5 |
| **Measurement location** | | | |
| Latitude (°N) | 40.050 | 40.045 | 40.050 |
| Longitude (°W) | 105.000 | 105.006 | 105.004 |
| Elevation (m ASL) | 1584 | 1587 | 1584 |







Table 2. Estimated uncertainties for TROPOZ, TOPAZ and LMOL ozone measurements within their measurable range
(see Table 1) for the 5 or 30-min integration time.

| Source | Uncertainty | |
|---|---|---|
| | 5-min integration | 30-min integration |
| Statistical error | <20% | <8% |
| Aerosol interference | <10% | |
| Interference by $SO_2$, $NO_2$, $O_2$ dimmer | <1.5% | |
| Differential Rayleigh scattering | <1% | |
| Total* | <22% | 13% |

*Total root-mean-square error.











**Table 3. Comparisons of the ozone column average measured by TROPOZ, TOPAZ, and LMOL.**

| Date | UTC time range | Altitude range (km) | Lidar | Number of the paired profiles | Mean ozone column average ($10^{16}$ molec·m$^{-3}$) | $1\sigma$ of the ozone column average ($10^{16}$ molec·m$^{-3}$) | Mean relative difference * | $1\sigma$ of the difference |
|---|---|---|---|---|---|---|---|---|
| 7/11/2014 | 1300 - 2135 | 0.6-2 | TROPOZ/ TOPAZ | 80 | 127.3/128.6 | 17.8/16.7 | -1.1% | 2.6% |
| 7/16/2014 | 1335 - 1730 | 0.9-2 | LMOL/T OPAZ | 28 | 98.1/102.0 | 13.1/13.0 | -3.8% | 2.9% |

* Equal to mean (A-B)/B for A/B in 'Lidar' column for all paired profiles.



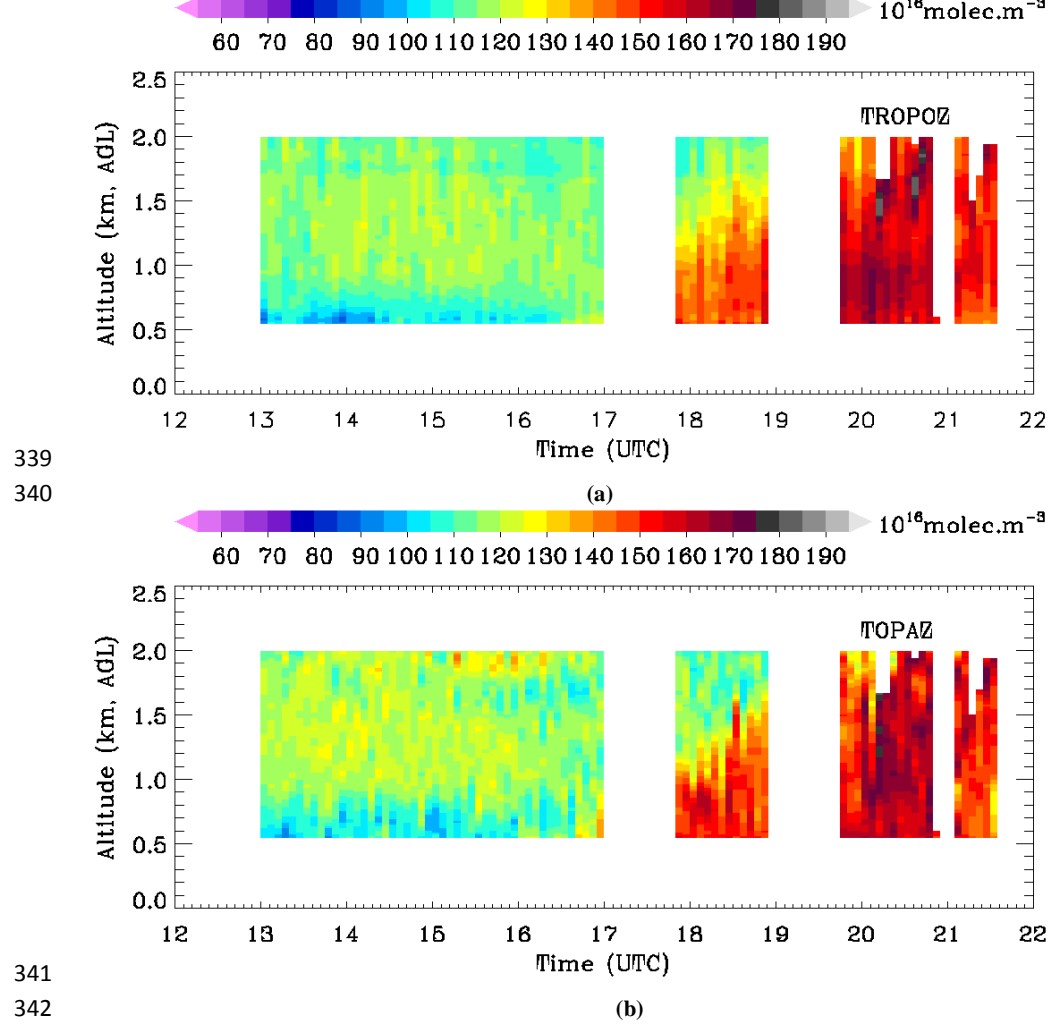




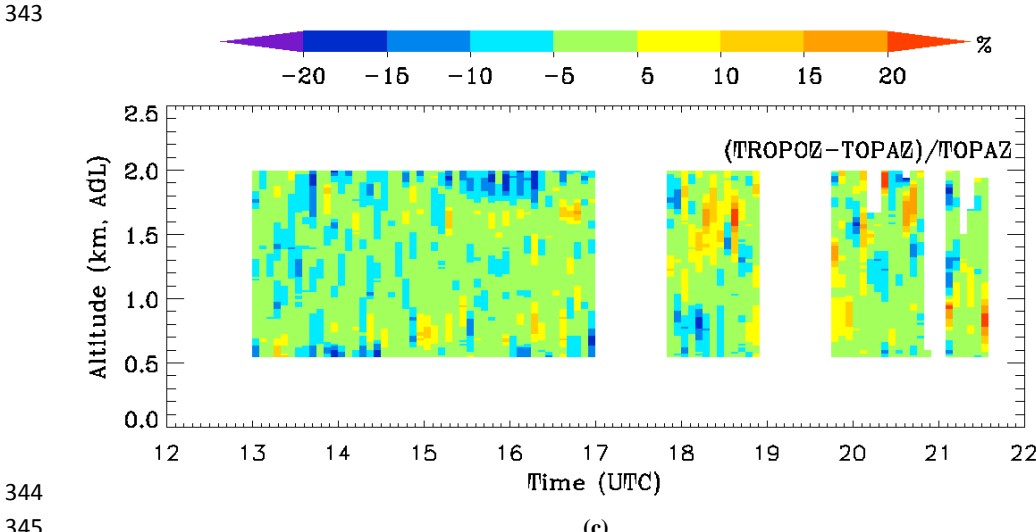

**(c)**

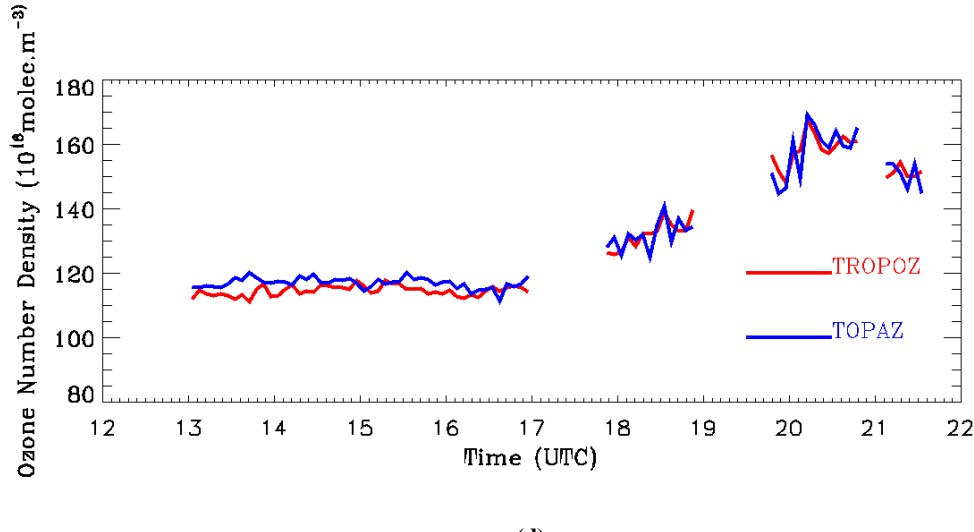

**(d)**

**Figure 1. Comparisons of ozone measured by TROPOZ and TOPAZ. (a) Ozone number densities measured by TROPOZ.**
**(b) Ozone number densities measured by TOPAZ. (c) Their relative percent differences, (TROPOZ-TOPAZ)/TOPAZ. (d)**
**Column averages measured by the TROPOZ and TOPAZ. TROPOZ measures 1.1±2.6% lower ozone column average**
**than TOPAZ.**



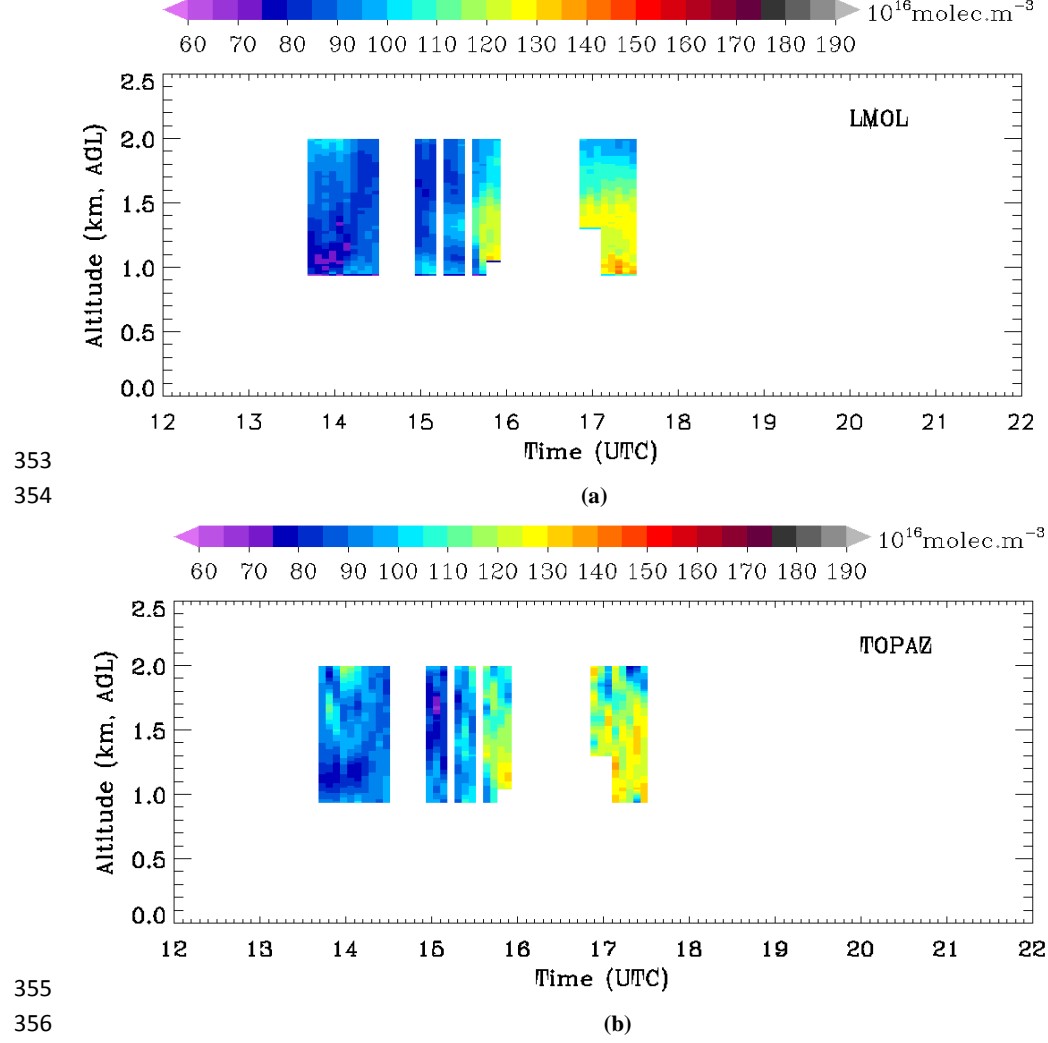


(a)


(b)





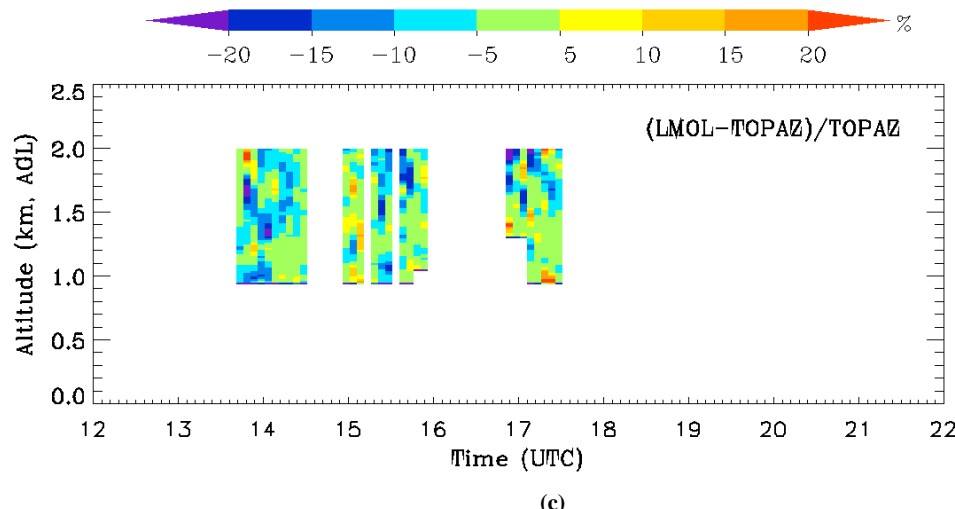

**(c)**

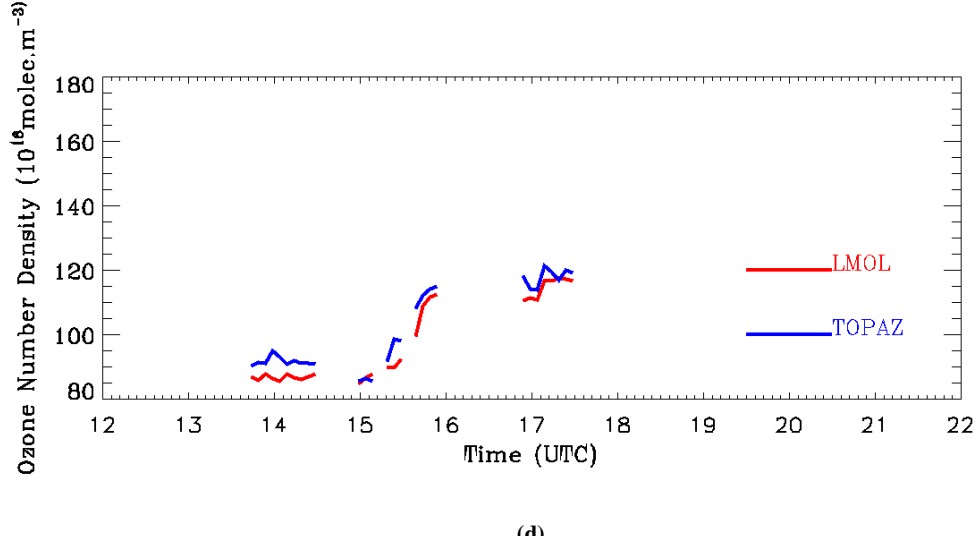

**(d)**

**Figure 2. Comparisons of ozone measured by LMOL and TOPAZ. (a) LMOL-measured ozone number densities. (b)**
**TOPAZ-measured ozone number densities. (c) Their relative percent differences, (LMOL-TOPAZ)/TOPAZ. (d) Column**
**averages measured by LMOL and TOPAZ. LMOL measures 3.8±2.9% lower ozone column average than TOPAZ.**






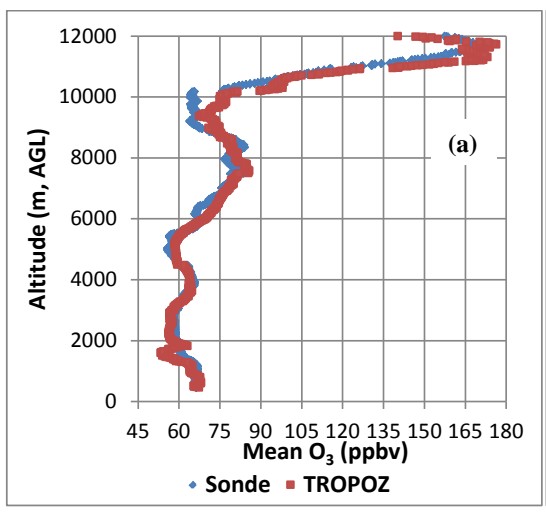

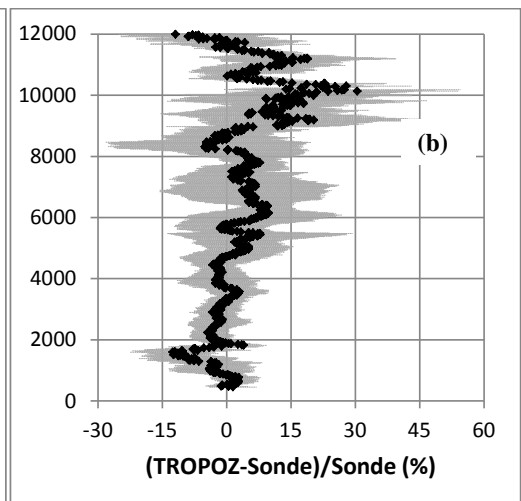

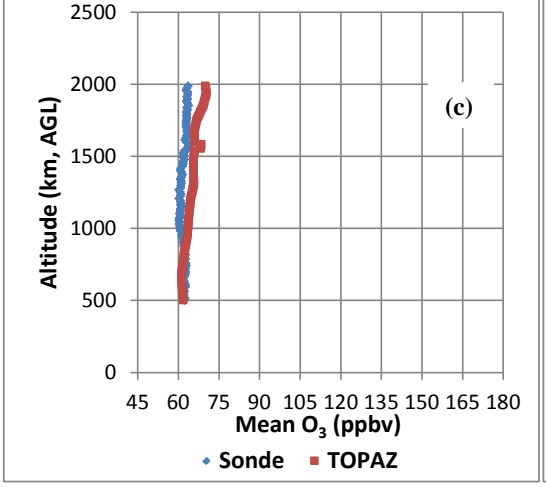

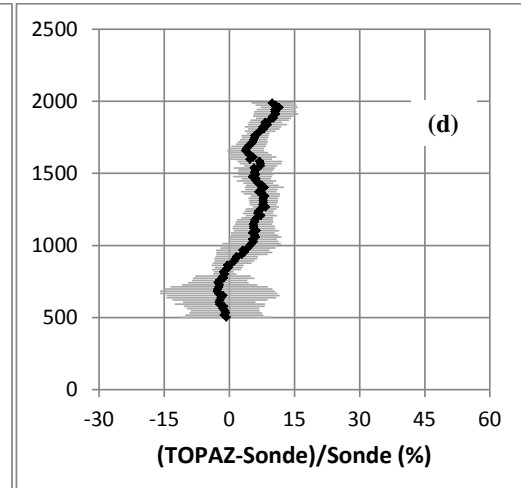






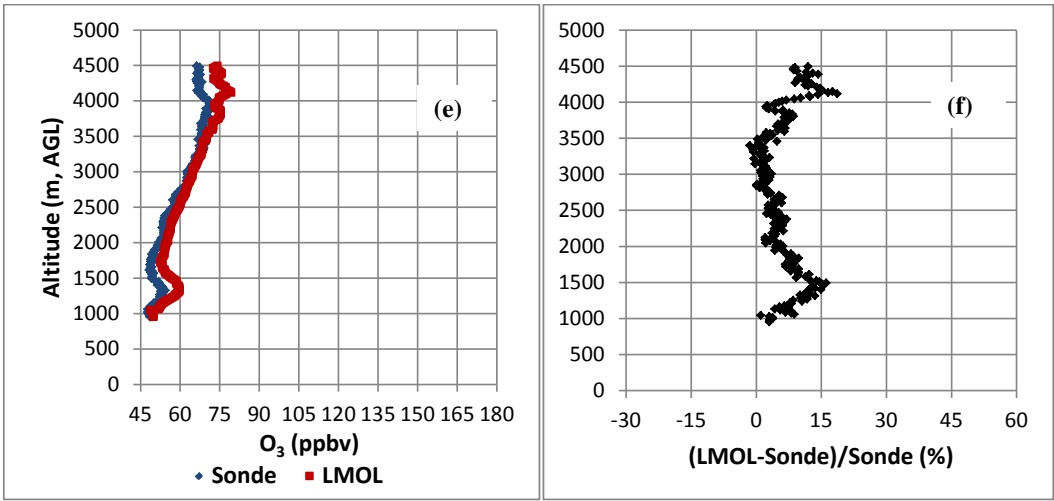


**Figure 3. Comparisons of lidar and ozonesonde measurements. (a) Average ozone profiles measured by TROPOZ and ozonesondes at Fort Collins, CO (11 pairs). (b) Mean relative difference between TROPOZ and ozonesondes as well as the 1-σ standard deviations. (c) Average ozone profiles measured by TOPAZ and ozonesondes at BAO Tower (7 pairs). (d) Mean relative difference between TOPAZ and ozonesondes. (e) Average ozone profiles measured by LMOL and ozonesonde at the BAO tower (1 pair). (f) Relative difference between LMOL and ozonesonde.**














**(e)**                                                            **(f)**

**Figure 4. Intercomparison between the lidar and P-3B measurements. (a) Average ozone profiles measured by TROPOZ**
**and P-3B at Fort Collins, CO (34 profiles). (b) Mean relative difference between TROPOZ and P-3B data as well as the 1-**
**σ standard deviation. (c) Average ozone profiles measured by TOPAZ and P-3B at the BAO Tower (29 profiles). (d) Mean**
**relative difference between TOPAZ and P-3B data. (e) Average ozone profiles measured by LMOL and P-3B at Golden,**
**CO (9 profiles). (f) Mean relative difference between LMOL and P-3B data.**

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
