# Peer review of "Quantifying TOLNet Ozone Lidar Accuracy during the 2014 2 DISCOVER-AQ and FRAPPÉ Campaigns"

_Atmospheric Measurement Techniques, 2017_

## Referee Comment (RC1) · Anonymous Referee #2 · 12 Jun 2017

The manuscript titled "Quantifying TOLNet Ozone Lidar Accuracy during the 2014 DISCOVER-AQ FRAPPE Campaigns" intercompares 3 different ozone lidars in the field as well as compares the lidar measurements to in situ sonde and aircraft measurements. The authors do a good job explaining the need for the scientific experiments and discuss the results in a clear and concise manner. Very few minor revisions can be made and are discussed below:

1. Line 159: How are the lidars selective for ozone as other compounds can absorb UV radiation at the wavelengths used here?

2. Line 265: "...overall positive bias..." implies that the ozonesondes are without error.

[Figure]

It is known that SO2 can interfere with the electrochemical ozone measurement. Were the ozonesonde data corrected for this artifact in any way? Do you have any reason to believe that SO2 impacted the measurement (e.g. through proximity to a coal-fired power plant)?

3. Section 3.2: When comparing the lidars with the P3, horizontal distances of up to 11 km were noted, yet horizontal differences were not discussed in this section. Since it is known the sondes do not travel directly upwards, differences between lidar and sondes could be due to real horizontal variability. Please discuss how this impacts the interpretation of your results.

---

## Referee Comment (RC2) · Anonymous Referee #1 · 16 Jun 2017

The manuscript reports on the intercomparison of three tropospheric ozone lidars, ECC ozone sondes and an aircraft-based chemoluminescence ozone instrument (P3B) during two field campaigns in Colorado in summer 2014. The goal is to investigate the accuracy of the lidars, that is to discover potential systematic biases, and to estimate and check their precision. This topic is well suited for Atmospheric Measurement Techniques. A thorough published characterization of system performance and accuracy certainly increases the value of these systems for tropospheric ozone research and monitoring.

While the manuscript presents substantial information about this intercomparison, I

feel that the necessary subsequent scientific analysis and evaluation is still lacking. Such analysis would be needed to draw firmer conclusions about system precision and potential biases. As it stands now, the results are rather vague, more like a report. What is missing, to me, is a thorough scientific analysis of the presented material. Also missing are clearer messages on the resulting biases and uncertainties. The current $\pm15\%$ given in the abstract is rather wide and generic, hardly meriting a new paper. I feel that with the information inherent in the manuscript much tighter and more specific uncertainties could be given, especially when aerosol interference does not seem to play a large role. I recommend to address the following major points, before the manuscript can be accepted for publication:

**1 General Comments and Questions**

1. Figs. 1d, 2d, and 3c,d indicate that the TOPAZ system generally reports higher ozone. Where is this bias coming from? Is it significant? Does it have something to do with the signal recording / background subtraction? Why do these error sources not appear in Table 2?

2. Fig. 4c-e, indicates a significant high bias of the P3B measurements. Given that TOPAZ (and possibly also LMOL, see Fig. 3e-f) seems to have a high bias against the sondes, the high bias of the P3B would be quite substantial. I think this possible bias needs to be investigated in more detail. It also needs to be reported in the abstract.

3. If significant, the potential biases in 1.) and 2.) need to be reported in the abstract. Or the authors have to clearly explain why they think these biases are not significant, and how they are covered by the different systems uncertainty budgets (e.g. in Table 1).

[Figure]

4. Apart from potential biases, the authors also need to verify the precision esti-
mates, e.g. those in Table 1. Since the statistical uncertainty (e.g. from pho-
ton counting statistics) is generally the largest uncertainty source for lidar ozone
profile measurements, it is very important to calculate and report that properly.
These estimated uncertainties then need to be checked using the statistics aris-
ing from multiple individual intercomparisons like the ones in this manuscript.
This important check, to me, is missing completely here.

5. For example, the scatter / standard deviations from Figs. 3b,d , and 4b,d,f need
to be compared to the estimated statistical uncertainty estimates available from
the lidars. This probably requires additional plots. The information can then be
used, on the one hand, to check the estimated lidar uncertainties, on the other
hand to check the estimated sonde and P3B precisions. To me, such checks are
a key component of an accuracy assessment. They are missing here.

6. Plots of average profile differences and their standard deviations should also be
generated for the comparisons in Figs. 1 and 2. They also need to be included in
the precision checks under 5., probably with additional plots and discussion.

7. In Figs. 4b,d,f, for example, it looks like the profile difference standard deviation
is of the order of $\pm 5\,\%$ $(1\sigma)$. This would indicate that the precision (repeatability)
of the lidar profiles is about 5%, assuming that the precision of the P3B profiles
is 1% as stated in 2.3 ($\sqrt{5^2 + 1^2} \approx \sqrt{25} = 5$). The precision of the 30 minute lidar
profiles would then be better than the 8% reported in Table 1 (and much better
than the 13% reported in Table 1). Similar considerations apply to the standard
deviations in Figs. 3b,d. Especially in Fig. 3b, the (expected) decrease of lidar
precision with altitude seems quite apparent to me, and this should be checked
against the lidar uncertainty estimates (e.g. from photon counting statistics).

**2 Detailed Remarks**

Line 1: Since the authors are only examining three of the many TOLNet lidars and certainly not TOLNet as a whole, I would suggest to move TOLNet after "2014" or after "FRAPPE".

Line 25: Replace "cross-instrument calibration" by "the network". The authors are not calibrating the lidars (I hope), they should be self-calibrating. Network uniformity is what the authors are really looking for.

Line 32: Drop "In terms of range resolving capability". I found this confusing, because there is really no investigation of consequences of the different and altitude dependent vertical resolutions of the lidars in the manuscript. This would be a whole separate issue, and therefore I would just drop this here.

Line 37: Replace "very good measurement accuracy for" by "that" and replace "making them" by "are". I am not sure that ±15% are "very good".

Line 44: Drop "high fidelity"? Is ±15% high fidelity? I don't think so. P3B claims 1% if biases are corrected.

Line 47: Swap "scientists" and "managers"? Or do the authors mean modeling and satellite managers?

Lines 56, 57: Replace "that ... their" by "of".

Line 59: Move "range resolution" after "operating ranges" in line 60. Range resolution is not really a hardware thing, and is much more determined by software.

Line 64: Add "can" before "form"?

Lines 67, 68: Drop "This particular study ... United States"? Is this relevant? Are the authors sure it is true? Was there no comparison, e.g. between TROPOZ and the Table Mountain tropospheric ozone DIAL?

Line 82: "selected" instead of "selective". Don't see how some sites would be more selective than others.

Line 82: Replace "profiles of ozone measurements" by "ozone profiles".

Line 93: Replace "lasers" by "pulses". Otherwise this would be a very expensive system indeed.

Line 102: Remove "zenith looking". As is now, this is confusing and contradictory.

Line 117: Add "s" after "measurement". Lines 117-127: This is a lot of text to say that, in the end, the system was just pointed to zenith. Shorten.

Lines 144,145: Drop "database" and "to calculate differential".

Line 152, 153: I do not understand what is done here. 5 points at 6 m hardware resolution would be 30 meters. 450 meters at 6 m hardware resolution would be 75 points. Explain / correct, also in Table 1.

Line 155: Please cite Leblanc et al. 2016 here. The authors should also include the other Leblanc et al. 2016 paper(s) on ozone profile uncertainties in the references. Also, the results here, i.e. range resolutions, ozone uncertainties and accuracies need to be properly put into the context of these papers, here and in other places in the text.

Line 168: Remove "non-standardized". Because it is so system specific, standardization is not really a criterion/ issue.

Line 174: Remove "maximum". What would that mean?

Line 179: Leblanc et al. 2016 on ozone profile uncertainties should be cited here, and should be put in context. Somewhere the authors should also mention that lidar uncertainty increases a lot with increasing altitude/ range.

Lines 181 to 191: To me, this is a bit backward. First the authors give the principle, then end results, and then the authors go back to the principle again. Rewrite / reorder.

Lines 108/109: These two citations should probably come before, on line 106 right after "observations".

Line 115: Replace "cloud interferences" by "clouds".

Line 223: Similar standard deviation could also be due to similar noise / precision / uncertainty. In fact, I think this seems to be the case from 13 to 17 UT, whereas similar variations seem to captured from 18 to 22 UT. Please reword.

Line 227: As mentioned above, please also show the mean and standard deviation profiles of these ozone differences. Same for Fig. 2.

Lines 265 to 271: I think this needs to be thought through much better. Are the sondes too low? Why would averaging time affect a bias? It should only affect the noise / significance. Same goes for SNR. Biases that are not resolvable/ not significant / within the uncertainty margins should not be discussed at all.

Lines 286 to 287: Why not the P3B? Figs. 4d and 4f look very similar. Many things point towards P3B being high. Same as the discussion of lidar sonde differences this discussion is to short. A lot more needs to be done / said here. See my major comments.

Table 2: Are these uncertainties $1\sigma$ or $2\sigma$?

Figs. 1d and 2d. Please plot (some/typical) error bars for these time series.

---

## Author Comment (AC1) · 25 Aug 2017

We thank both reviewers for their significant time and constructive comments which improve the value of this paper. We have accepted Reviewers' suggestions and made following important changes 1) added profiles of estimated uncertainties for ozone lidar measurements in Figure 3 and 4 which could be compared to the 1-sigma of the differences between lidar and ozonesonde (or P3-B) and also added their corresponding description; 2) Added the uncertainty budget for each lidar system in Table 2 for a more rigorous characterization of measurement uncertainties. Following are the detailed answers for Reviewers' specific questions. (The line numbers refer to the version with

tracked changes.)

Reviewer 1 The manuscript titled "Quantifying TOLNet Ozone Lidar Accuracy during the 2014 DISCOVER-AQ FRAPPE Campaigns" intercompares 3 different ozone lidars in the field as well as compares the lidar measurements to in situ sonde and aircraft measurements. The authors do a good job explaining the need for the scientific experiments and discuss the results in a clear and concise manner. Very few minor revisions can be made and are discussed below: 1. Line 159: How are the lidars selective for ozone as other compounds can absorb UV radiation at the wavelengths used here?

In principle, the two wavelengths are selected at which the ozone absorptions are significantly different while the extinction for other gases and aerosols are small enough. The wavelengths of TOPAZ and LMOL are tunable and have been optimized for minimizing the differential scattering/absorption from other species (primarily SO2). But, these two systems are relatively complicated and hard to maintain. The TROPOZ lasers are more straightforward and easier to maintain, but their wavelengths are fixed (289, 299nm). This fixed-wavelength pair 289-299 has larger interference from SO2 than other two systems and we will explain the details in Question 3. We have presented the error budget for all lidar systems in Section2.1 and Table 2. We have also stated that the corrections for differential Rayleigh scattering and aerosol interference have been regularly applied for all DIAL retrievals.

2. Line 265: "...overall positive bias..." implies that the ozonesondes are without error. We change to say "...all three TOLNet lidars measured higher ozone than ozonesondes with mean ozone column differences of 2.9 % for TROPOZ, 4.4% for TOPAZ, and 6.2 % for LMOL (based on a single profile comparison)".

3. It is known that SO2 can interfere with the electrochemical ozone measurement. Were the ozonesonde data corrected for this artifact in any way? Do you have any reason to believe that SO2 impacted the measurement (e.g. through proximity to a coal-fired power plant)?

We have added the uncertainty discussion due to SO2 for both ozonesondes and the lidars. There's no any sign for the lidar, sonde measurements which were contaminated by SO2. This can be known by comparing TROPOZ data to TOPAZ and LMOL which have minimum SO2 interference. There's unlikely high SO2 emission around Boulder, CO at this time. But we still state this possibility as a generally potential error source in the ozonesonde instrument description as following in Section 2.2: "It has been reported that the ECC sondes suffer interference from SO2 (Flentje et al., 2010) with 1-ppb SO2 being registered as -1-ppb ozone (Schenkel and Broder, 1982). Elevated SO2 can be a concern for lidar-ozonesonde intercomparison for some lidar wavelengths (e.g., 289-299 nm) because of the opposite signs of the measurement error arising from SO2 for lidar and ozonesondes. However, this is not an issue for this study since we did not find any noticeable interference from SO2 in either lidar or ozonesonde data." In terms of lidar measurement, the SO2 absorption cross section in the Hartley band varies a lot and brings large uncertainty for the calculation. SO2 is also a potential interfering specie for 289-299 pair and we have added more description about this error source in Section 2.1.5. Table 2 is modified as well.

4. Section 3.2: When comparing the lidars with the P3, horizontal distances of up to 11 km were noted, yet horizontal differences were not discussed in this section. Since it is known the sondes do not travel directly upwards, differences between lidar and sondes could be due to real horizontal variability. Please discuss how this impacts the interpretation of your results.

To address this question, we added following description in the 1st paragraph in Section 3.2: "Ozonesondes and lidars do not sample exactly the same atmospheric volume because the sondes typically drift horizontally. Therefore, discrepancies between the lidar and sonde observations may be in part due to real atmospheric differences. The horizontal displacement of the sonde usually increases with altitude, so the distance between sonde and lidar is normally larger in the free troposphere than in the PBL. However, horizontal ozone gradients tend to be smaller in the free troposphere than in the PBL, which typically keeps atmospheric differences rather small despite the increased displacement of the sonde."

Reviewer 2 The manuscript reports on the intercomparison of three tropospheric ozone lidars, ECC ozone sondes and an aircraft-based chemoluminescence ozone instrument (P3B) during two field campaigns in Colorado in summer 2014. The goal is to investigate the accuracy of the lidars, that is to discover potential systematic biases, and to estimate and check their precision. This topic is well suited for Atmospheric Measurement Techniques. A thorough published characterization of system performance and accuracy certainly increases the value of these systems for tropospheric ozone research and monitoring. While the manuscript presents substantial information about this intercomparison, I feel that the necessary subsequent scientific analysis and evaluation is still lacking. Such analysis would be needed to draw firmer conclusions about system precision and potential biases. As it stands now, the results are rather vague, more like a report. What is missing, to me, is a thorough scientific analysis of the presented material. Also missing are clearer messages on the resulting biases and uncertainties. The current 15% given in the abstract is rather wide and generic, hardly meriting a new paper. I feel that with the information inherent in the manuscript much tighter and more specific uncertainties could be given, especially when aerosol interference does not seem to play a large role. I recommend to address the following major points, before the manuscript can be accepted for publication:

To address these concerns, we have enhanced the analysis of the lidar measurement uncertainties for each system and added calculated lidar uncertainties in Figure 3 and 4 compared to the one sigma of the differences between the lidars and validation instruments. The expected measure precisions for all lidars are consistent with the measured precisions.

General Comments and Questions 1. Figs. 1d, 2d, and 3c,d indicate that the TOPAZ system generally reports higher ozone. Where is this bias coming from? Is it significant? Does it have something to do with the signal recording / background subtraction?

Why do these error sources not appear in Table 2?

We think the measurement differences between TOPAZ and the other two lidar are acceptable although these differences are noticeable. TOPAZ measured higher than other lidars and sondes, but, measures lower than P3-B in average. This means TOPAZ does not persistently measure higher than other instruments. At L257 in Section 3.1, we explain the possible causes for the differences as: "This small, but statistically significant ozone column difference could be due to errors in the background and saturation corrections, or biases introduced by the merging of signals or ozone retrievals from different instrument channels."

2. Fig. 4c-e, indicates a significant high bias of the P3B measurements. Given that TOPAZ (and possibly also LMOL, see Fig. 3e-f) seems to have a high bias against the sondes, the high bias of the P3B would be quite substantial. I think this possible bias needs to be investigated in more detail. It also needs to be reported in the abstract.

After further investigation, we still think the P3B measurement is correct and has a measurement precision as claimed, 5%. The comparisons of TOPAZ-P3B and LMOL-P3B indeed don't look perfect in Figure 4. But, the biases are mostly within expected. We have added the expected total uncertainties in Figure 4, as the reviewer suggested, including the 5% P-3B uncertainty to account for the potential errors from P3-B.

In the last paragraph of Section 3, we provide the explanation for these biases as "The differences between the three lidars and the P-3B are not significantly correlated suggesting that these biases were not caused by the P-3B ozone instrument. These differences could at least in part be caused by the lidar systematic errors mentioned in Section 2.1.5, but could also reflect horizontal ozone variability across the P-3B spirals, which were up to 22 km in diameter."

3. If significant, the potential biases in 1.) and 2.) need to be reported in the abstract. Or the authors have to clearly explain why they think these biases are not significant, and how they are covered by the different systems uncertainty budgets (e.g. in Table

1).

We have enhanced the discussion of the error budget for each system in Table 2 and Section 2.1.5. We have adopted a more standard classification for error sources and added the uncertainties due to background correction and saturation correction. We have also added the expected uncertainties in Figure 3 and 4 to compare with the actual differences between lidar and sondes (or P3-B). The differences between different instruments generally smaller than expected uncertainties suggesting our understanding of these error sources are correct.

4. Apart from potential biases, the authors also need to verify the precision estimates, e.g. those in Table 1. Since the statistical uncertainty (e.g. from photon counting statistics) is generally the largest uncertainty source for lidar ozone profile measurements, it is very important to calculate and report that properly. These estimated uncertainties then need to be checked using the statistics arising from multiple individual intercomparisons like the ones in this manuscript. This important check, to me, is missing completely here.

We have accepted the suggestions and plotted the expected uncertainties (green lines in Figure 3 and 4) compare to the actual 1-sigma standard deviations. We have also provided more discussions on these changes. For example, we added in Section 3.2, "The green lines in Figure 3 (b) represent the expected total measurement uncertainties including the lidar measurement uncertainties for a 30-min integration time (also see Table 2) and a 10% constant uncertainty for ozonesondes. The purple lines represent the $1$-$\sigma$ standard deviations of the mean differences, which can be compared to the combined precision of lidar (i.e., statistical uncertainty) and ozonesonde (5%). The $1$-$\sigma$ standard deviation increases from about 10% in the lower troposphere to about 20% in the upper troposphere as a result of increasing lidar statistical uncertainties with altitude. Below 9 km, the $1$-$\sigma$ standard deviations of the mean differences are mostly located within the range of the expected uncertainties. In particular, the lidar-sonde differences around 0.5 km are significantly less than the expected uncertainties
suggesting that the detection and counting systems of TROPOZ performed better than anticipated."

5. For example, the scatter / standard deviations from Figs. 3b,d , and 4b,d,f need to be compared to the estimated statistical uncertainty estimates available from the lidars. This probably requires additional plots. The information can then be used, on the one hand, to check the estimated lidar uncertainties, on the other hand to check the estimated sonde and P3B precisions. To me, such checks are a key component of an accuracy assessment. They are missing here.

As mentioned above, we have added.

6. Plots of average profile differences and their standard deviations should also be generated for the comparisons in Figs. 1 and 2. They also need to be included in the precision checks under 5., probably with additional plots and discussion.

We believe we have provided enough discussions for the lidar comparisons including four plots in both Figure 1 and 2, and Table 3. These figures and discussions have covered comparisons for individual grids, column average, standard deviations. As mentioned earlier, we show the expected total uncertainty in both Figure 3 and 4 for all systems.

7. In Figs. 4b,d,f, for example, it looks like the profile difference standard deviation is of the order of $\pm 5$ % ($1\sigma$). This would indicate that the precision (repeatability) of the lidar profiles is about 5%, assuming that the precision of the P3B profiles is 1% as stated in 2.3 ($\sqrt{(5^2+1^2)}\approx\sqrt{25}=5$). The precision of the 30 minute lidar profiles would then be better than the 8% reported in Table 1 (and much better than the 13% reported in Table 1) <means Table 2>. Similar considerations apply to the standard deviations in Figs. 3b,d. Especially in Fig. 3b, the (expected) decrease of lidar precision with altitude seems quite apparent to me, and this should be checked against the lidar uncertainty estimates (e.g. from photon counting statistics).

[Figure]

We agree with the reviewer's method to check the consistency between the actual bias and estimated uncertainties. We have broken down the numbers in Table 2 for separate systems. Table 2 reports the maximum uncertainties within each lidar's measurement range. The highest measurement altitude for TROPOZ is higher than 12 km. But the highest altitude shown in Figure 4(b) is only 4 km due to P3B's flying altitude. So the 1-sigmas of TROPOZ-P3B look smaller than "the maximum" in Table 2. We have added the discussions on the comparison of 1-sigma and lidar precision. The results show the actual precision is consistent with our estimates in Table 2. For example, at L304 we add "The 1-$\sigma$ standard deviation of the mean differences (purple lines) is about 5% which is close to the combined precision of TOPAZ and ozonesondes (about 6%). 1-$\sigma$ of the mean differences stays almost entirely within the expected uncertainties indicative of a proper estimate of the lidar measurement uncertainties for TOPAZ in Table 2." At L340, we add "The 1-$\sigma$ standard deviation of the LMOL-P3-B relative differences is mostly between 5% and 8% and is consistent with their combined precision (6%). The 1-$\sigma$ of the mean differences for both TOPAZ and LMOL (purple lines in Figure 4 d, f) stays within the expected uncertainty (green lines) except for the bottom altitudes."

Detail comments: Line 1: Since the authors are only examining three of the many TOLNet lidars and certainly not TOLNet as a whole, I would suggest to move TOLNet after "2014" or after "FRAPPE".

We agree that we are examining only three of the TOLNet lidars. But, because "TOL-Net" is not the same category as DISCOVER-AQ or FRAPPE and rephrasing as the reviewer suggested may also cause confusion, we would still keep the current title.

Line 25: Replace "cross-instrument calibration" by "the network". The authors are not calibrating the lidars (I hope), they should be self-calibrating. Network uniformity is what the authors are really looking for.

Replaced.

Line 32: Drop "In terms of range resolving capability". I found this confusing, because there is really no investigation of consequences of the different and altitude dependent vertical resolutions of the lidars in the manuscript. This would be a whole separate issue, and therefore I would just drop this here.

Deleted

Line 37: Replace "very good measurement accuracy for" by "that" and replace "making them" by "are". I am not sure that _15% are "very good".

Changed as suggested.

Line 44: Drop "high fidelity"? Is _15% high fidelity? I don't think so. P3B claims 1% if biases are corrected.

Removed.

Line 47: Swap "scientists" and "managers"? Or do the authors mean modeling and satellite managers?

Changed to "scientists and managers within the air quality, modeling, and satellite communities"

Lines 56, 57: Replace "that . . . their" by "of".

Replaced.

Line 59: Move "range resolution" after "operating ranges" in line 60. Range resolution is not really a hardware thing, and is much more determined by software.

We agree the range resolution doesn't 100% belong to a hardware category because it could be adjusted when the software of the counting system is designed this way. But, the range resolution is associated with the capability of the counting system and so is an important parameter of hardware. So, we choose not to move "range resolution" here.

Line 64: Add "can" before "form"?

Added.

Lines 67, 68: Drop "This particular study . . . United States"? Is this relevant? Are the authors sure it is true? Was there no comparison, e.g. between TROPOZ and the Table Mountain tropospheric ozone DIAL?

This sentence has been removed.

Line 82: "selected" instead of "selective". Don't see how some sites would be more selective than others.

Changed as suggested.

Line 82: Replace "profiles of ozone measurements" by "ozone profiles".

Changed it to "measurements of ozone profiles".

Line 93: Replace "lasers" by "pulses". Otherwise this would be a very expensive system indeed.

Replaced.

Line 102: Remove "zenith looking". As is now, this is confusing and contradictory.

Changed.

Line 117: Add "s" after "measurement".

Added

Lines 117-127: This is a lot of text to say that, in the end, the system was just pointed to zenith. Shorten.

Shortened.

Lines 144,145: Drop "database" and "to calculate differential".

Dropped.

Line 152, 153: I do not understand what is done here. 5 points at 6 m hardware resolution would be 30 meters. 450 meters at 6 m hardware resolution would be 75 points. Explain / correct, also in Table 1.

Changed the sentence to "The TOPAZ group averaged lidar signal over 90 m and, then, smoothed the derivative of the logarithm of the signal ratios with a five-point least-square fitting in a 450-m interval."

Line 155: Please cite Leblanc et al. 2016 here. The authors should also include the other Leblanc et al. 2016 paper(s) on ozone profile uncertainties in the references. Also, the results here, i.e. range resolutions, ozone uncertainties and accuracies need to be properly put into the context of these papers, here and in other places in the text.

We have added [Leblanc et al., 2016b] and cited papers by Leblanc [2016a, b] at L63, 154, 188. We have brought these values into the discussions, especially the uncertainties.

Line 168: Remove "non-standardized". Because it is so system specific, standardization is not really a criterion/ issue.

Removed.

Line 174: Remove "maximum". What would that mean?

To avoid confusion, we rephrase the sentence as "The statistical uncertainty, often referred to as measurement precision, generally increases with range due to decreasing SNR and is different for the three lidars due to their different laser power, telescope sizes, and measurement ranges."

Line 179: Leblanc et al. 2016 on ozone profile uncertainties should be cited here, and should be put in context. Somewhere the authors should also mention that lidar uncertainty increases a lot with increasing altitude/ range.

[Figure]

The Leblanc et al., 2016b paper has been cited here. Yes, we have said so at L179 in the error budget section and other places, and also stated the statistical uncertainty was range dependent in the footnote of Table 2.

Lines 181 to 191: To me, this is a bit backward. First the authors give the principle, then end results, and then the authors go back to the principle again. Rewrite / reorder.

As suggested, we have deleted the repeating words and reordered this paragraph.

Lines 108/109: These two citations should probably come before, on line 106 right after "observations". <Means line number "208 and 209">

We agree and have moved them as suggested.

Line 115: Replace "cloud interferences" by "clouds".

Replaced.

Line 223: Similar standard deviation could also be due to similar noise / precision /uncertainty. In fact, I think this seems to be the case from 13 to 17 UT, whereas similar variations seem to captured from 18 to 22 UT. Please reword.

We agree the sigma could come from uncertainty, primarily statistical uncertainty which is a random noise. However, the ozone variations in Figure 1 (a) and (b) don't look like random noises. So, we add "(also see Figure 1 a and b)".

Line 227: As mentioned above, please also show the mean and standard deviation profiles of these ozone differences. Same for Fig. 2.

Please see the answer for General comment 6.

Lines 265 to 271: I think this needs to be thought through much better. Are the sondes too low? Why would averaging time affect a bias? It should only affect the noise / significance. Same goes for SNR. Biases that are not resolvable/ not significant / within the uncertainty margins should not be discussed at all.

We have deleted the citation of the Gaudel et al. 2015 paper because it is not comparable to this study. The Gaudel paper compares seasonally averaged lidar and sonde O3 profiles that were not necessarily taken at the same time. Then, we modify this paragraph as "In summary, all three TOLNet lidars measured higher ozone than ozonesondes with mean ozone column differences of 2.9 % for TROPOZ, 4.4% for TOPAZ, and 6.2 % for LMOL (based on a single profile comparison). . The differences between the two types of instruments and the standard deviations are mostly less than the expected uncertainties. The largest bias occurs at far-range altitudes as expected and is primarily associated with the high statistical errors arising from low SNR. The increased bias at near-range altitudes could be associated with various factors, primarily the aerosol correction and the merging of the signals or ozone retrievals from different optical or altitude channels."

Lines 286 to 287: Why not the P3B? Figs. 4d and 4f look very similar. Many things point towards P3B being high. Same as the discussion of lidar sonde differences this discussion is to short. A lot more needs to be done / said here. See my major comments.

If P3B has a significant measurement bias, we expect all three lidars to have similar measurement differences relative to P3-B. But now, only two of them look similar. So, we don't think P3-B had significant measurement issue.

Table 2: Are these uncertainties 1_ or 2_?

Added $1\sigma$ in the title of Table 2.

Figs. 1d and 2d. Please plot (some/typical) error bars for these time series.

Added the error bars and their discussions.

Please also note the supplement to this comment:
https://www.atmos-meas-tech-discuss.net/amt-2017-106/amt-2017-106-AC1- supplement.pdf

**Supplement:**

[revised manuscript text omitted]